# Tracking Spatiotemporal Patterns of Rwanda's Electrification Using Multi-Temporal VIIRS Nighttime Light Imagery

**Yuanxi Ru** [1], **Xi Li** [2,*] and **Wubetu Anley Belay** [2]

1   School of Remote Sensing and Information Engineering, Wuhan University, Wuhan 430079, China
2   State Key Laboratory of Information Engineering in Surveying, Mapping and Remote Sensing, Wuhan University, Wuhan 430079, China
*   Correspondence: lixi@whu.edu.cn; Tel.: +86-180-6241-1350

**Abstract:** After recovering from the Rwanda Genocide in the last century, Rwanda is experiencing rapid economic growth and urban expansion. With increasing demand for electricity and a strong desire to achieve the Sustainable Development Goals (SDGs), it is significant to further investigate the electrification progress in Rwanda. This study analyzes the characteristics of electrification in Rwanda from 2012 to 2020 using VIIRS nighttime light imagery. Firstly, by analysis of the nighttime light change patterns on a national scale, we find that the electrification in Rwanda is seriously unbalanced, as electrification progress in Kigali is much faster than that in the rest of the country. Secondly, there is a common phenomenon where power grid expansion in Rwanda fails to keep pace with rapid urbanization, especially in areas with an inadequate electricity infrastructure foundation. Quantitatively, original electricity infrastructure level shows a positive impact on the grid access of new settlements, with an $R^2$ value of 0.695 in the linear regression. In addition, new settlements inside the urban boundary tend to achieve more extensive grid access compared to those outside the boundary. Finally, the grid access rates are calculated on multi-spatial scales. By comparing the calculated results with the official electricity access rate data, we analyze the development of off-grid access in Rwanda. The results imply that, since 2016, off-grid access has rapidly developed in Rwanda, especially in the rural areas, playing an important role in achieving the SDGs.

**Keywords:** Rwanda; nighttime light; electrification; VIIRS; power grid





## 1. Introduction

The Rwanda Genocide in 1994 was a devastating blow to Rwanda's social and economic development. However, after reflection and efforts, Rwanda is now one of the countries with the fastest economic growth in Africa. Electricity access plays a vital role in accelerating economic development and improving the standards of living [1]. Rwanda has a lower starting point for electrification compared to other countries in Africa [2]. In 2000, only 6.2% of the population had access to electricity [3]. The government of Rwanda has a target for electrifying the whole country with 52% on grid and 48% off grid by 2024 [2], attaching great importance to the promotion of electrification. Evaluation of the expected electrification progress is important for Rwanda and can help the national government and international community to adjust the electrification strategy for this country.

The United Nations' Sustainable Development Goal 7 (SDG 7) aims at ensuring access to affordable, reliable, sustainable, and modern energy for all by 2030. Access to electricity services (SDG targets 7.1.1 and 7.A and 7.B) is a key priority under this goal [4], but achieving it in Rwanda still faces a lot of challenges. Spatially explicit information on the electrification progress in Rwanda is needed to identify where efforts should be focused to achieve the most benefits. Nevertheless, available electricity access data for Rwanda can, at best, be found mostly at the national level, for example, the World Bank provides the annual electricity access rates data in Rwanda from 1997 to 2020. The electricity access rate

in 2012 was 17.5%, while the electricity access rate in 2020 was 46.6% [3]. The data provided by the World Bank provide little spatial information. In fact, electricity access data for subnational areas in Rwanda are usually publicly unavailable [5]. Therefore, the spatial patterns and regional disparity of Rwanda's delivery of electricity are unclear. In addition, traditional electricity access measurement relies predominantly on expensive and inefficient statistical surveys which are labor intensive and rapidly outdated [6]. More spatially detailed information and efficient data measurement are essential for clearly understanding the electrification status of Rwanda and tracking its progress toward the SDGs.

Nighttime light (NTL) data, which are a common source of remote sensing data, have great potential to monitor measures of human-related socioeconomic activity, such as GDP [7–9], population [10–12], built-up area density [13], house vacancy rate [14], urbanization monitoring [15–19], electric power consumption [20–22], etc. NTL has long been recognized as a valuable indicator of the availability and use of electricity around the world. Studies show that nighttime light output strongly correlates with electricity generating capacity at the regional and national levels [23–25]. Compared to traditional statistical survey data, NTL data can provide a unique perspective for the visualization and analysis of the spatial distribution of electricity access in a consistent, efficient, and low-cost manner [26]. By monitoring nighttime brightness, nighttime satellite imagery helps to evaluate the electrification status in Rwanda at multiple spatiotemporal scales. Previous studies have shown that combining nighttime light and human settlement datasets can help to track the progress of electrification even at a local scale and estimate multi-spatial electricity access levels [6]. For example, Min et al. detected rural electrification in Vietnam and Africa using DMSP-OLS nighttime light data [25,27]. Falchetta et al. derived multi-dimensional estimates of electricity access over space and time by processing high-resolution population distribution maps, satellite-measured nighttime light, and settlement locations for sub-Saharan Africa [28].

In addition to tracking the electrification progress, it is also important to analyze the regional disparity in electrification. The presence of excessive regional electrification inequality is a hindrance to sustainable development and affects stability in society [29]. Understanding the regional disparity in electrification helps to identify the backward areas which need the most attention. Therefore, electrification can be highly promoted, and progress towards SDG 7 can be accelerated. Previous studies have used the nighttime light satellite imagery to analyze regional inequality [29–32]. For example, Elvidge et al. developed the Night Light Development Index (NLDI) to provide a spatial depiction of differences in development within countries [30]. Singhal et al. employed the association between nighttime light and economic activities for India at the sub-state or district level and calculated regional income inequality using Gini coefficients [29]. Xu et al. used the NLDI to measure the regional inequality of public services in Mainland China on multiple scales [32].

In many African countries, power grid expansion has been proved to be the greatest way to increase electrification [2]. However, the conventional electricity supply system provided by grid expansion is economically unfeasible for people living in rural areas who are far from the existing power grid [33–37]. Recently, off-grid access has served as an alternative with lower costs and higher efficiency and has the potential to deliver equally technically and economically feasible electrification solutions for rural households and those without access to the grid [2]. The government of Rwanda has supported off-grid electricity generation to increase electricity access and inexpensive systems [2]. Now, Rwanda's off-grid electricity market is dominated by the Solar Home System (SHS), providing 550,000 people with access [2]. However, off-grid access still has its limitations. Studies show that SHS is limited in scope as basic energy needs, such as energy requirements for cooking, cannot be met [38]. Besides the low power capacity, the energy conversion efficiency is low and can easily be affected by deposition of dust and other elements on the panels [39]. Therefore, compared to the power grid, off-grid access provides a less steady electricity supply.

To evaluate the electrification status in Rwanda with spatially explicit information and efficient measurement, this study analyzed the electrification progress in Rwanda using the nighttime light data with ancillary data and reference data. Firstly, we analyzed the general distribution of nighttime light in Rwanda at multiple spatiotemporal scales, which provides reference for understanding the changing trend and regional disparity of the electrification in Rwanda. Secondly, the lit ratio of settlements was calculated in order to evaluate whether electricity infrastructure construction has kept pace with the expansion of settlements in Rwanda. Thirdly, we examined the correlations between the new settlements' access to grid and their original electricity infrastructure level, as well as their spatial location, to find out the possible influencing factors of power grid expansion. Finally, we calculated the grid access rates on multi-spatial scales and compared our calculated results with the official electricity access rates data. By comparison, this study analyzed the development of off-grid access as an alternative electricity supply in Rwanda.

## 2. Materials and Methods

### 2.1. Study Area and Data

#### 2.1.1. Study Area

Rwanda is located in East Africa at approximately two degrees below the equator. It has borders with Burundi in the south, Democratic Republic of Congo in the west, Tanzania in the east, and Uganda in the north. The international, provincial, and district boundaries of Rwanda are derived from the Database of Global Administrative Areas (GADM) (http://www.gadm.org/, accessed on 26 October 2021). As shown in Figure 1, Rwanda is divided into four provinces, Eastern Province, Western Province, Northern Province, and Southern Province, and one city with provincial status: Kigali. Further, Rwanda is divided into 30 districts.

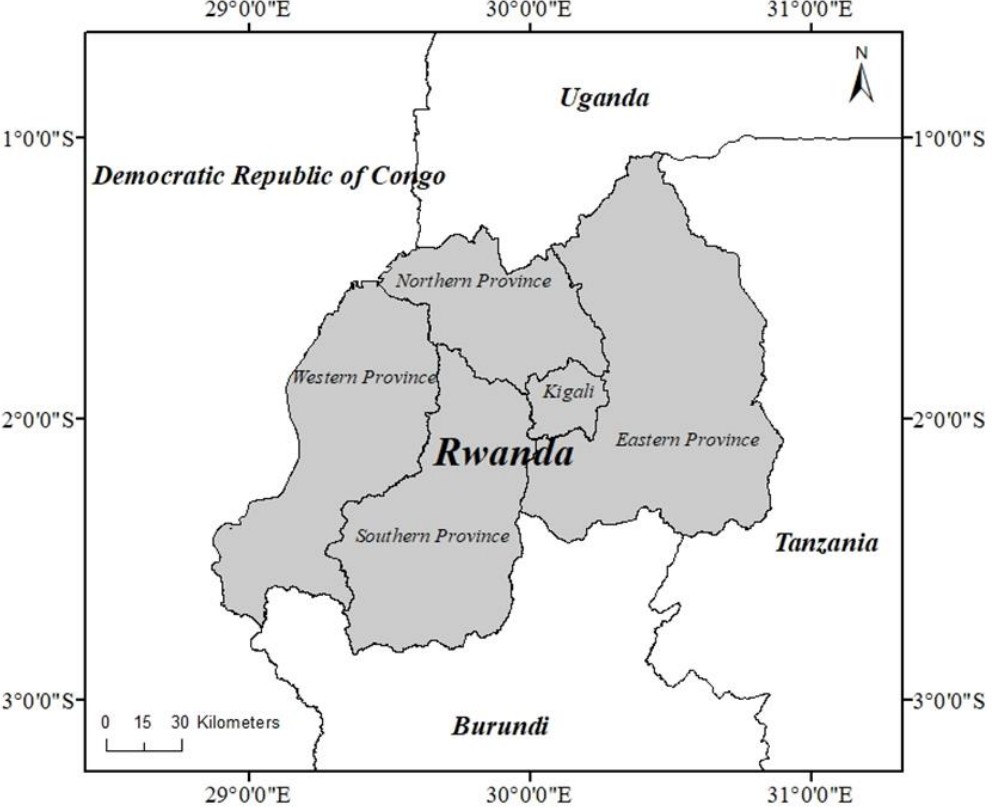

**Figure 1.** The administrative map of Rwanda, including its four provinces, Kigali, and neighboring countries.

### 2.1.2. Nighttime Light Satellite Imagery

The nighttime light data collected by the Day/Night Band (DNB) sensors of the Visible Infrared Imaging Radiometer Suite (VIIRS) on board the Suomi National Polar-Orbiting Partnership (S-NPP) satellite platform were used in this study. Recently, the National Aeronautics and Space Administration (NASA) has developed the Black Marble (VNP46) suite, which plays a vital role in research on urbanization [40], disaster-related power outages [41], proxies for economic activity [42], and the changing conditions in human settlements (i.e., urban energy access, migration, and disaster impact and recovery) [43]. This study employed the all-angle snow-free layer in the Black Marble annual nighttime light product (VNP46A4) from 2012 to 2020 (https://blackmarble.gsfc.nasa.gov/#product, accessed on 9 February 2022). The all-angle snow-free layer provides annual composite products generated from all daily observations during the snow-free period [44]. The spatial resolution of the product is 15 arc second [44]. Figure 2 shows the nighttime light composite imagery of Rwanda in 2012, 2016, and 2020.

Compared to other nighttime light products, the superiority of the Black Marble product suite lies in its temporal resolution and its overall data quality. It provides high-quality nighttime light data based on cloud-free, atmospheric-, terrain-, vegetation-, snow-, lunar-, and stray-light-corrected nighttime VIIRS DNB radiances, which minimize the influence of extraneous artifacts and biases [44]. The corrected nighttime radiances, resulting in the superior retrieval of nighttime light on short time scales and a reduction in background noise, enable quantitative analyses of daily, seasonal, and annual variations [40]. Additionally, the Black Marble suite achieves substantial sensitivity enhancement of low-lit structures, which made it more suitable to be used in this study [45,46].

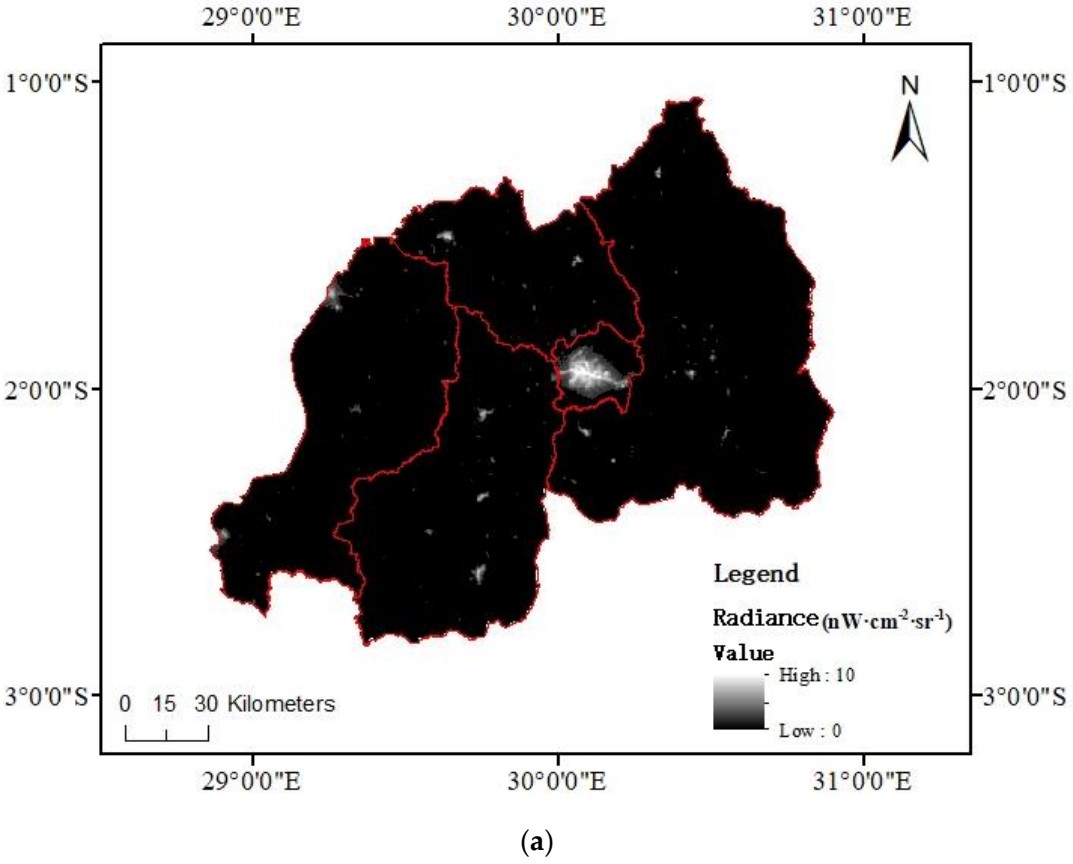

(**a**)

**Figure 2.** *Cont.*

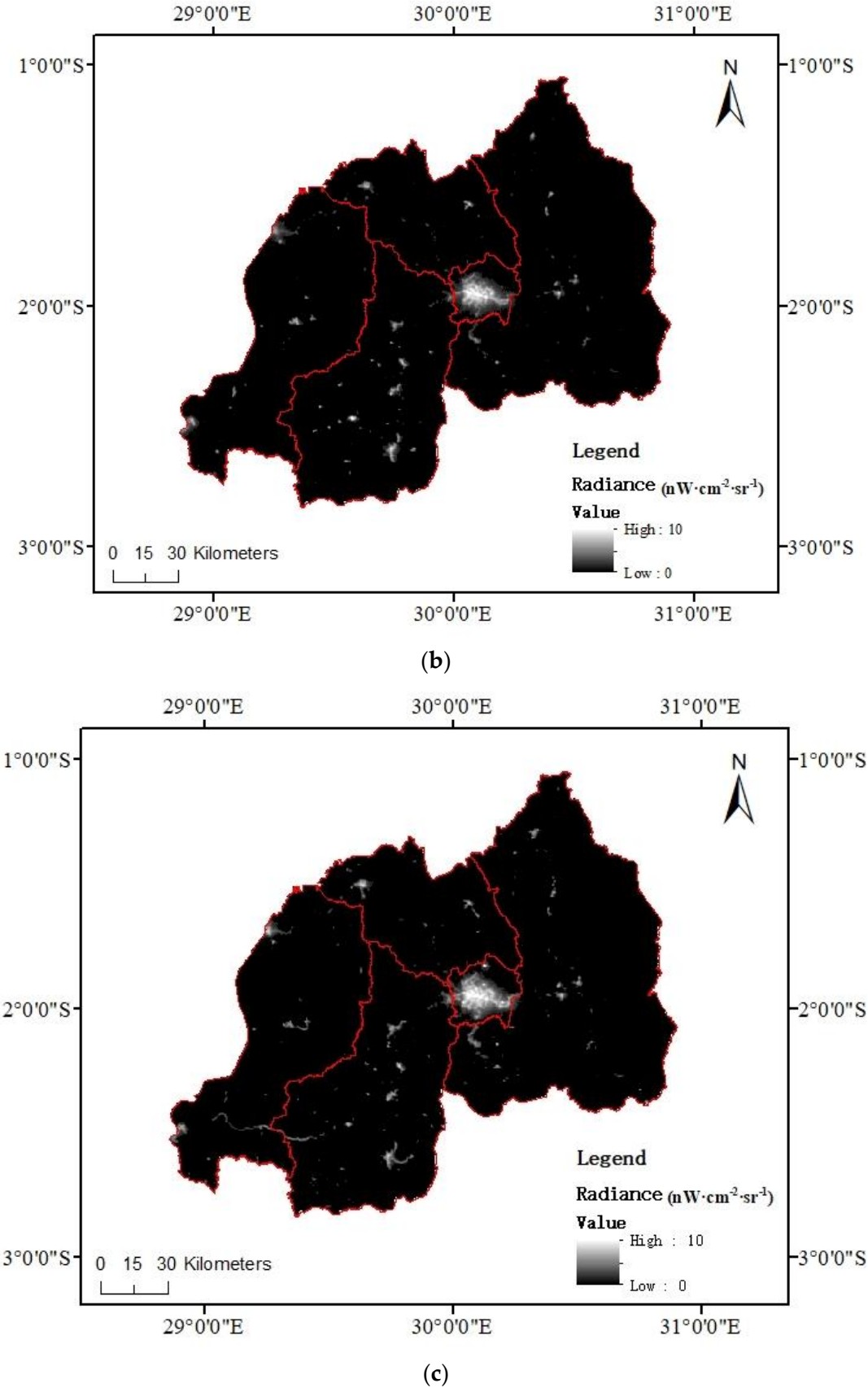

**Figure 2.** The nighttime light composite imagery of Rwanda: (**a**) 2012; (**b**) 2016; (**c**) 2020.

### 2.1.3. Ancillary Data

To analyze the electrification progress of settlements and calculate the grid access rates in Rwanda, four types of ancillary data source were used in this study:

(1)　Human settlement location data. The settlement location data of Rwanda were obtained from the World Settlement Footprint (WSF) dataset (https://geoservice.dlr.de/web/maps/eoc:guf:3857, accessed on 5 February 2022). The WSF dataset is a 10 m resolution binary mask outlining the extent of human settlements globally. In this study, we resampled the settlement data to the same spatial resolution as the nighttime light imagery (i.e., 15 arc second). Figure 3 shows the distribution of settlements in Rwanda.

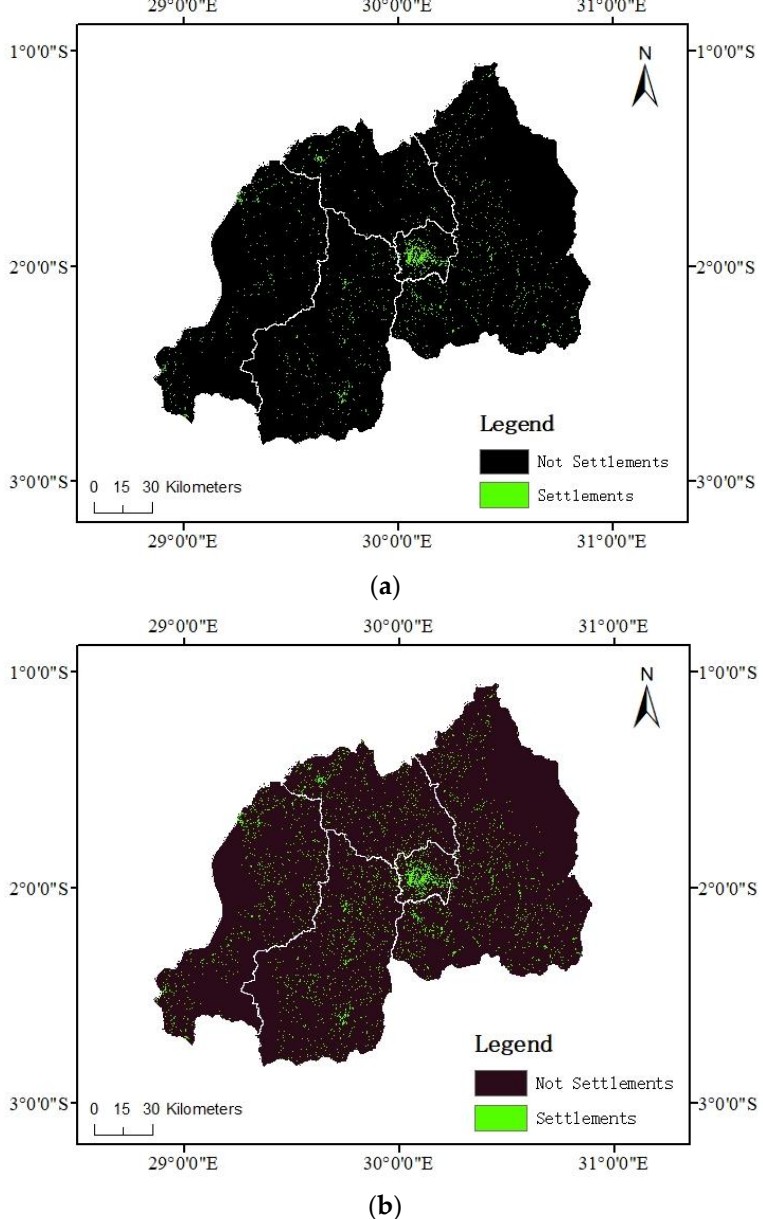

**Figure 3.** The human settlement map in Rwanda: (**a**) 2015; (**b**) 2019.

(2)　Urban agglomeration boundary data. The boundaries of the major cities in Rwanda were obtained from the Africapolis dataset (https://africapolis.org/en, accessed on 2 March 2022). Africapolis is the only international database that systematically

includes all small urban agglomerations with more than 10,000 residents. It comprises 7600 African settlements, 97% of which are home to fewer than 300,000 people.

(3) Population density data. The population density distribution of Rwanda was obtained from the WorldPop United Nations adjusted population density datasets (https://www.worldpop.org/, accessed on 12 February 2022). The WorldPop population density dataset maps the global population at a high resolution, which provides annual population density data estimated at the level of cells with a resolution of 30 arc s (approximately 1 km at the equator). The units are number of people per square kilometer based on a country's total population adjusted to match the corresponding, official United Nations population estimates.

(4) Urban–rural settlement classification data. The Global Human Settlement Layers (GHSL) dataset was mapped based on the Landsat imagery to show the global built-up areas and population distribution from 1975 to 2014 [47]. The GHS Settlement Model (GHS-SMOD) is the urban–rural settlement classification model adopted by the GHSL [48]. The GHS-SMOD data have been generated by integrating the GHSL built-up areas and GHSL population data [48]. The GHS-SMOD dataset in 2015 (https://ghslsys.jrc.ec.europa.eu/download.php?ds=smod, accessed on 3 March 2022) was selected for dividing the urban and rural areas in Rwanda. The settlements were classified as urban, rural, or not inhabited [6]. Figure 4 shows the classification of the settlements in Rwanda.

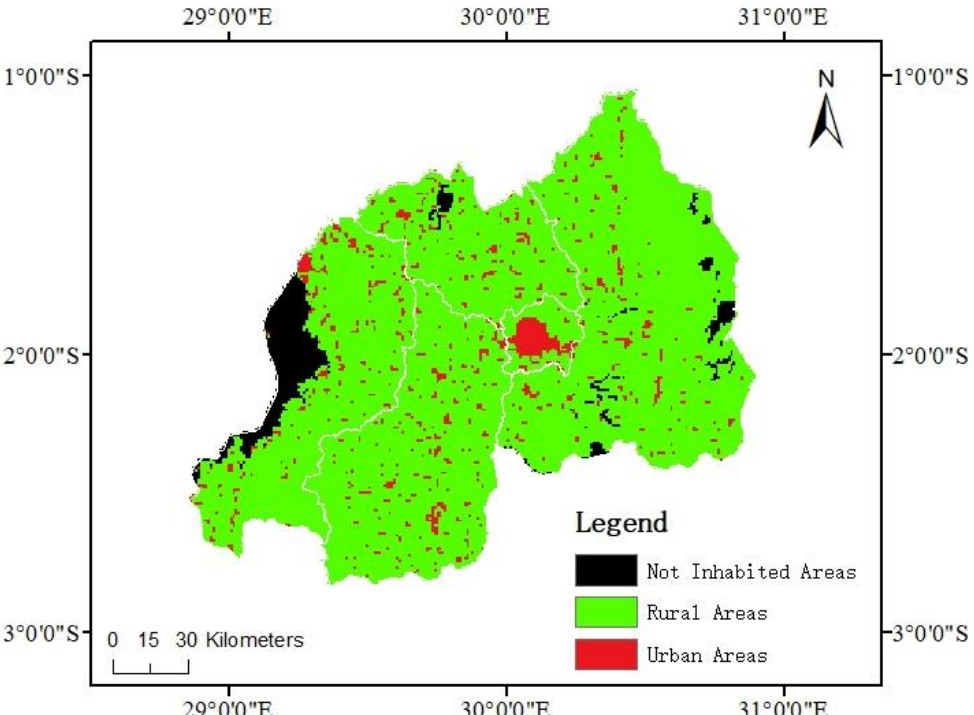

**Figure 4.** The urban, rural, and not inhabited areas in Rwanda.

2.1.4. Reference Data

In order to compare our calculated grid access rates with the official electricity access rate data, this study used the electricity access rate data of Rwanda provided by the World Bank and the Demographic and Health Surveys (DHS) as reference data.

The World Bank works in every major area of development. It provides a wide array of financial products and technical assistance and helps countries to share and apply innovative knowledge and solutions to the challenges they face [49]. The electricity access rate data of Rwanda from the World Bank were employed as a reference for comparison with our calculated national grid access rates [3].

The DHS project is a regular survey on living conditions and health-related data that is conducted across many countries [50]. It uses a survey instrument which contains

questions at the individual level but also the household level [50]. The DHS provides electricity access rate data at the provincial level and at the urban–rural level in Rwanda for a limited number of years (https://dhsprogram.com, accessed on 14 February 2022) and was used in this study for comparison and reference.

### 2.2. Methods

The goal of this study was to analyze the electrification progress in Rwanda on multi-spatial scales. From the nighttime light imagery, we obtained the distribution of nighttime light in Rwanda and analyzed its general spatial patterns. In addition, the lit ratio was proposed to track the electrification progress of the settlements in Rwanda. Then, we further explored the possible influencing factors of power grid access for new settlements and analyzed the correlation between them. Finally, the grid access rates were calculated on multi-spatial scales in order to estimate the electrification status of Rwanda.

In Rwanda, electricity access consists of on-grid and off-grid access [1]. Previous studies showed that the city light mainly comes from the light on the roads [51,52]. Yet off-grid access tends to achieve only a limited supply of electricity to meet daily household needs and is unable to provide electricity supply for streetlights and other public lighting [2,53]. Additionally, off-grid access equipment usually does not have an electricity storage function and can only provide electricity supply for a limited time [54]. Therefore, the nighttime light of off-grid-access regions is weak and unstable, which makes it hard for satellites to record. As a result, the lit areas we extracted were only the areas connected to the power grid.

### 2.2.1. Analysis of Nighttime Light Patterns in Rwanda

In order to analyze the changing trend of nighttime light in Rwanda, we calculated the sum of light (SOL) in Rwanda from 2012 to 2020. The SOL is calculated based on the nighttime light radiance value of all pixels in a region. The formula is as follows:

$$SOL = \sum_{i=1}^{N} radiance_i, \tag{1}$$

where $SOL$ denotes the sum of light, $radiance_i$ denotes the nighttime light radiance value of the pixel $i$, and $N$ denotes the counts of the pixels in the region.

To analyze the regional disparity of the distribution of the nighttime light in Rwanda, we also computed the SOL and the NTL per capita of each province based on the corresponding annual nighttime light imagery. The NTL per capita is calculated by dividing the SOL by the total population of the corresponding region. Additionally, we also calculated the SOL and the NTL per capita of the urban and rural areas. By overlaying the urban–rural binary image of Rwanda with the nighttime light imagery and the population density data, the SOL and the total population in the urban and rural areas were calculated and then the NTL per capita was derived. The formula of the NTL per capita is as follows:

$$NTL_{per\ capita} = \frac{SOL}{P_{total}}, \tag{2}$$

where $NTL_{per\ capita}$ denotes the nighttime light per capita, $SOL$ denotes the sum of light, and $P_{total}$ denotes the total population in the region.

To further detect the changes in nighttime light, we compared the nighttime light in 2012 and 2020 by extracting the areas where the pixel value was zero in 2012 but positive in 2020, which were the newly lit areas. By doing this, the areas which were dark in 2012 but appeared lit in 2020 could be observed. By the analysis of the change detection image, we can better understand the spatial patterns of the key areas of the electrification progress in Rwanda. These areas have made a huge leap in electrification progress and deserve more attention.

2.2.2. Analysis of Grid Access of Settlements in Rwanda

This study defined the lit ratio of settlements (abbreviated as lit ratio in the following) as the proportion of the lit settlements compared to the total settlements. Lit settlements are settlements with positive nighttime light radiance. The formula of the lit ratio is as follows:

$$R_{lit} = \frac{A_{lit}}{A_{total}}, \tag{3}$$

where $R_{lit}$ denotes the lit ratio of the settlements in Rwanda, and $A_{lit}$ and $A_{total}$ denote the surface area of the lit settlements and the total settlements, respectively.

To analyze the expansion of the settlements in Rwanda, we calculated the change rate of the settlements from 2015 to 2019. The calculation is carried out using the following formula:

$$R_{settlement} = \frac{A_{new}}{A_{original}}, \tag{4}$$

where $R_{settlement}$ denotes the change rate of settlements in Rwanda, and $A_{new}$ and $A_{original}$ denote the surface area of the new settlements and original settlements in Rwanda, respectively. The calculated change rate of the settlements was 98.62%, indicating that Rwanda is experiencing rapid expansion of the settlements.

Under the rapid expansion of settlements, it is significant to analyze the electrification progress of the settlements in Rwanda. As the WSF dataset provides settlement data for Rwanda in 2015 and 2019, we calculated the lit ratio of Rwanda in these two years. Additionally, to analyze the regional disparity of the electrification progress of the settlements in Rwanda, the lit ratio in each province in 2015 and 2019 was calculated.

The impact of two potential influencing factors, the original electricity infrastructure level and the spatial location, on the lit ratio was analyzed. The way to calculate the two factors was as follows.

(1) The Original Electricity Infrastructure Level

In areas that have a certain electricity infrastructure foundation, the distance between the newly built power grid and the existing power grid is shorter, which may reduce the cost and difficulty of grid expansion [55,56]. This study analyzed the correlation between the original electricity infrastructure level in a region and the grid access of its new settlements.

As studies show that nighttime light strongly correlates with electricity generating capacity at the regional and national levels [23–25], the original electricity infrastructure level and the grid access of new settlements can be reflected by the initial lit ratio and the lit ratio of new settlements, respectively. In other words, a larger lit ratio can represent a higher level of electricity infrastructure in an area. The lit ratio of new settlements refers to the proportion of the surface area of new settlements in light compared to that of the total new settlements. The formula of the lit ratio of the new settlements is as follows:

$$R_{new\ lit} = \frac{A_{new\ lit}}{A_{new\ total}}, \tag{5}$$

where $R_{new\ lit}$ denotes the lit ratio of the new settlements, and $A_{new\ lit}$ and $A_{new\ total}$ denote the surface area of the new settlements in light and the total new settlements, respectively.

In order to analyze the correlation between the original electricity infrastructure level in a region and the grid access of its new settlements, we calculated the initial lit ratio (lit ratio of settlements in 2015) and the lit ratio of new settlements from 2015 to 2019, corresponding to the 30 districts in Rwanda. Taking the initial lit ratio as the independent variable and the lit ratio of the new settlements as the dependent variable, this study employed linear regression to analyze the correlation between the two variables. The formula of the linear regression model is as follows:

$$R_{new\ lit} = aR_{lit} + b, \tag{6}$$

where $R_{new\ lit}$ denotes the lit ratio of the new settlements, $R_{lit}$ denotes the lit ratio of settlements in 2015, and *a* and *b* denote the regression coefficient and intercept, respectively.

(2)　The spatial location

This study also analyzed the impact of new settlements' spatial location on their grid access. The analysis can be divided into the following five steps:

(1) Generate the urban boundary by establishing buffer zones with a radius of 1 km based on the urban agglomeration vector data of the major cities in Rwanda which were obtained from the Africapolis dataset;

(2) Subtract the settlement data of 2015 from those of 2019 to obtain the new settlement image;

(3) Overlay the nighttime light imagery with the new settlement image to obtain the lit new settlements in Rwanda;

(4) Extract the new settlements and the lit new settlements inside and outside the urban boundary using the generated urban boundary. Figure 5 is the schematic diagram of new settlements inside and outside the urban boundary;

(5) Calculate and compare the lit ratio of the new settlements inside and outside the urban boundary.

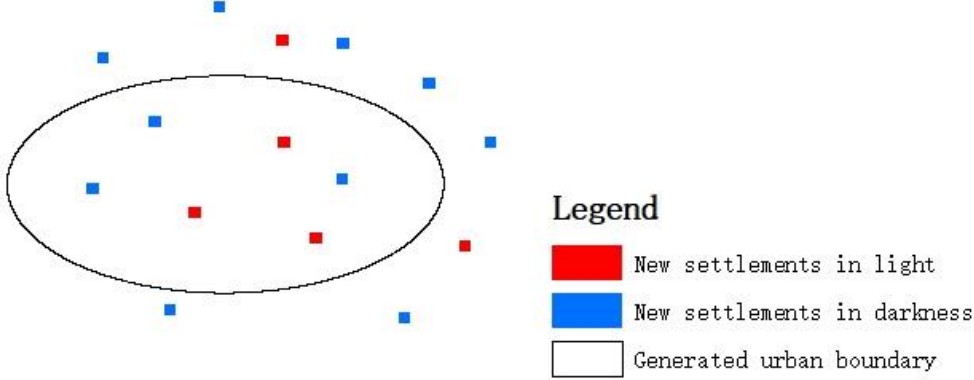

**Figure 5.** The schematic diagram of new settlements inside and outside the urban boundary. The red areas show the new settlements with positive nighttime light radiance, while the blue areas show the new settlements in darkness.

Further, to find the possible impact of the disparity in the spatial location of the new settlements in different provinces on the regional disparity of the lit ratio, we calculated and compared the change rates of the settlements inside and outside the urban boundary at the provincial level.

2.2.3. Calculation of the Grid Access Rates

Nighttime light has already been used as a proxy for electricity access [57–59]. As clarified at the beginning of Section 2.2, the extracted lit areas in this study were only the areas connected to the power grid, and the calculated access rate using the nighttime light imagery was the grid access rate. The grid access rate refers to the proportion of the population with access to the grid compared to the total population. The formula of the grid access rate is as follows:

$$R_{grid} = \frac{P_{grid}}{P_{total}}, \tag{7}$$

where $R_{grid}$ denotes the grid access rate, and $P_{grid}$ and $P_{total}$ denote the population with access to grid and the total population, respectively.

The grid access rate was calculated by the following four steps:

(1) By overlaying the population density data with the nighttime light imagery, the populated pixels with positive radiance were extracted and identified as 'achieved grid access', whereas the populated pixels with zero radiance were classified as 'not

achieved grid access'. This strategy has been successfully used for creating a binary mask of whether light is present or not in a previous study [57]. As a result, a binary image of whether grid access has been achieved in populated areas was obtained;

(2)    By overlaying the binary image of grid access with the population density data, a thematic map showing population with access to grid was obtained;

(3)    The total population and the population with access to grid were added up;

(4)    By dividing the population with access to grid and the total population counts, the grid access rate was calculated.

Furthermore, to obtain the grid access rates on multi-spatial scales, we calculated the grid access rates at the provincial level and at the urban–rural level.

## 3. Results

### 3.1. Nighttime Light Dynamics in Rwanda

This study tracked the temporal trend in the SOL of Rwanda from 2012 to 2020, which is shown in Figure 6. Although the SOL has shown some slight fluctuation, the nighttime light in Rwanda has continuously increased since 2012. The SOL in 2020 was about twice as much as that in 2012, which indicates that electrification in Rwanda is progressing rapidly.

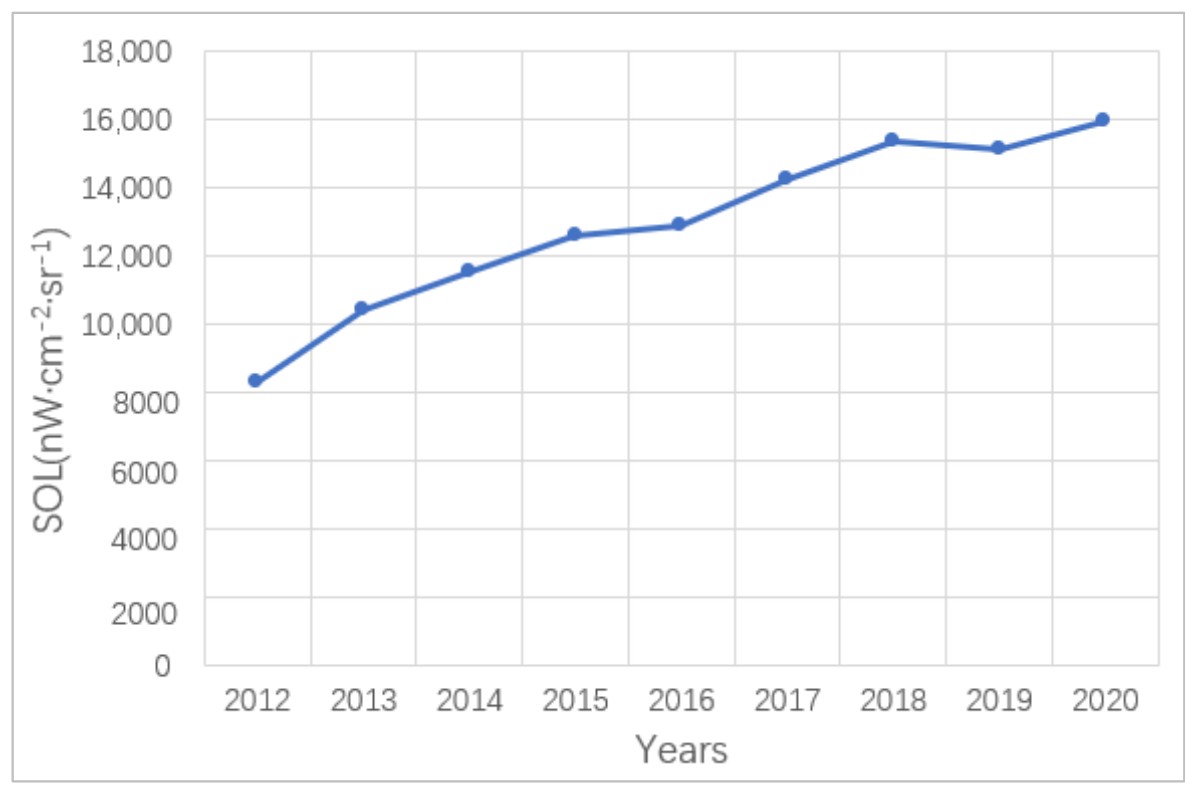

**Figure 6.** SOL of Rwanda from 2012 to 2020.

To analyze the temporal trends of the SOL at the provincial level and compare the SOL of different provinces, the SOL of each province in Rwanda was calculated. As shown in Figure 7, the SOL in all provinces showed a growing trend. However, the results also showed the overwhelming dominance of Kigali with regard to SOL, which reflects the extreme imbalance in the distribution of nighttime light in Rwanda's provinces. Each province has nearly the same area, but the SOL of Kigali was even more than the sum of those of the other four provinces.

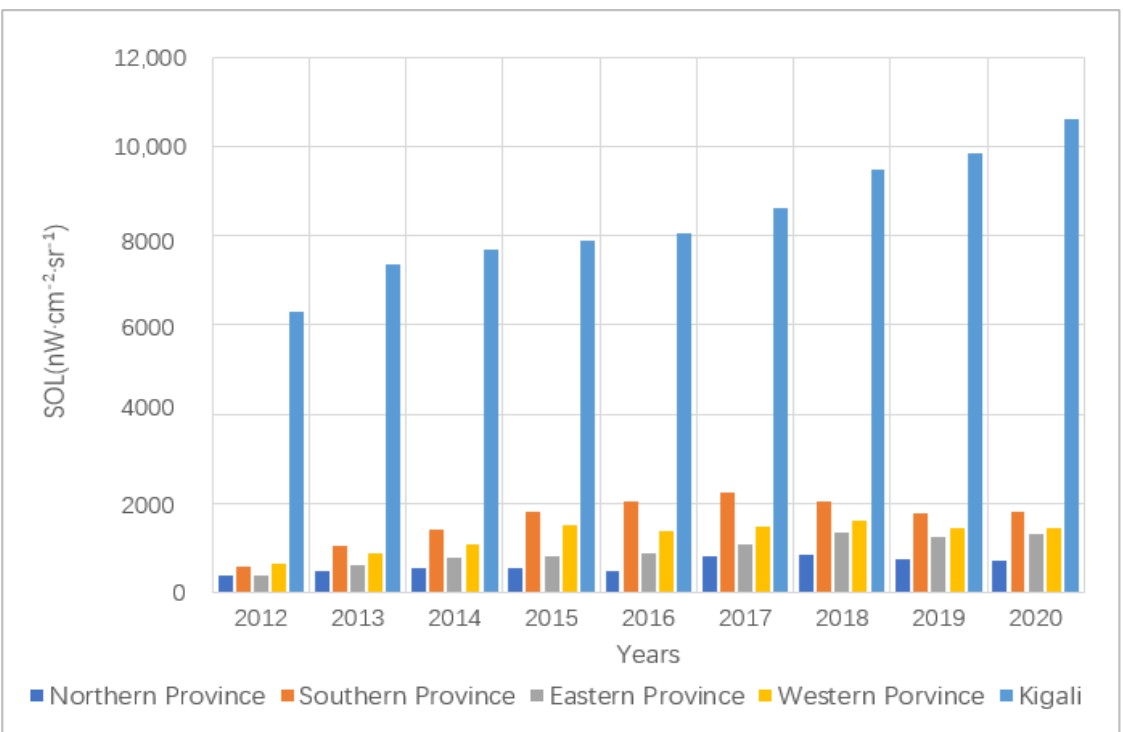

**Figure 7.** SOL of each province in Rwanda from 2012 to 2020.

Further, we calculated the NTL per capita of each province; the results are shown in Figure 8. The NTL per capita of each province showed a more obvious disparity. The NTL per capita in Kigali was almost 20 times greater than that in the other four provinces. The findings further illustrate that there is a serious imbalance in the delivery of electricity in Rwanda, which presents a stumbling block for the country's efforts to achieve the SDGs.

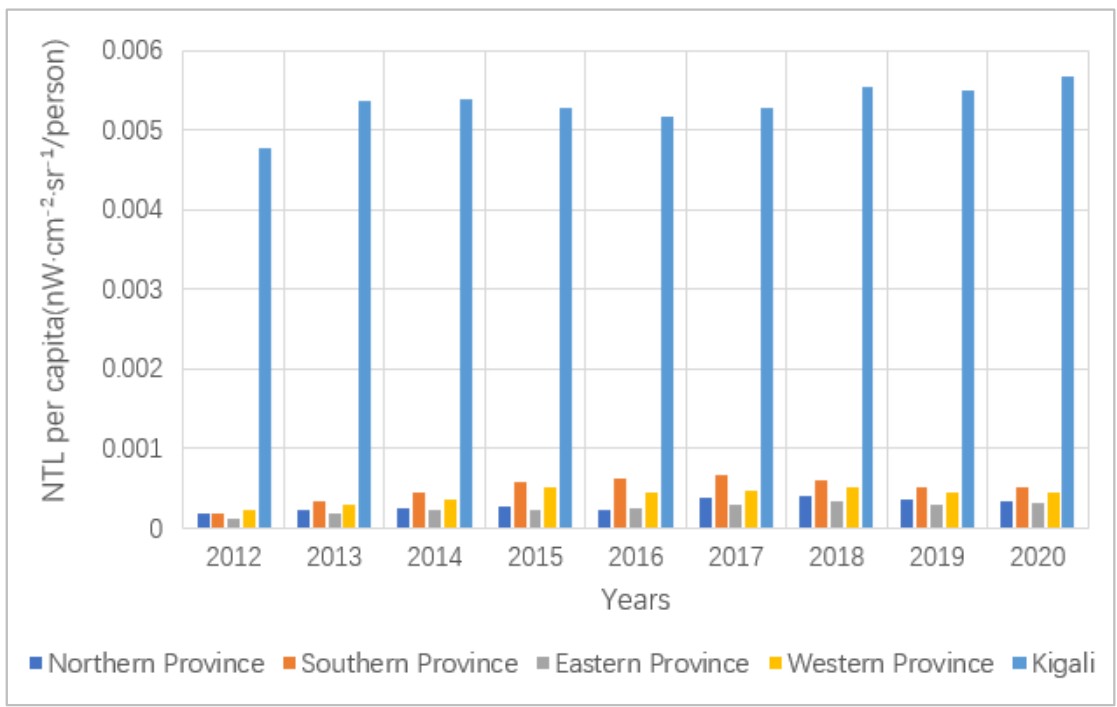

**Figure 8.** NTL per capita of each province in Rwanda from 2012 to 2020.

In this study, we also calculated the SOL of the urban and rural areas in Rwanda. As shown in Figure 9, the SOL of the urban areas was much higher than that of the rural areas. In 2020, the SOL of the rural areas was 908 nW·cm$^{-2}$·sr$^{-1}$, while the SOL of the urban areas was 2871.4 nW·cm$^{-2}$·sr$^{-1}$, which was about triple that of the rural areas.

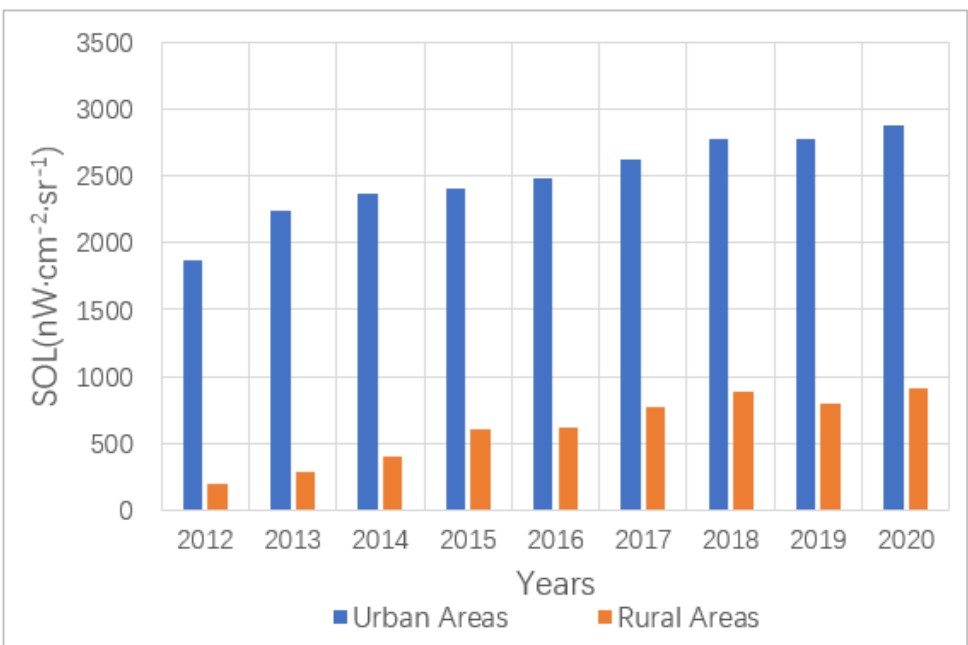

**Figure 9.** SOL of the urban and rural areas in Rwanda from 2012 to 2020.

In addition, we calculated the NTL per capita of the urban and rural areas, and the results are shown in Figure 10. From the results, it can be seen that the NTL per capita of the urban areas was much larger than that of the rural areas, showing a greater gap than the SOL. The results show a serious imbalance in the distribution of the nighttime light in the urban and rural areas. Yet the SOL and the NTL per capita of both areas have been growing steadily since 2012, and the gap between the two is narrowing year by year, which indicates that the imbalance problem has been improved in recent years. In 2012, the NTL per capita of the urban areas was about 50 times greater than that of the rural areas. However, by 2020, the gap between them had narrowed to about 15 times.

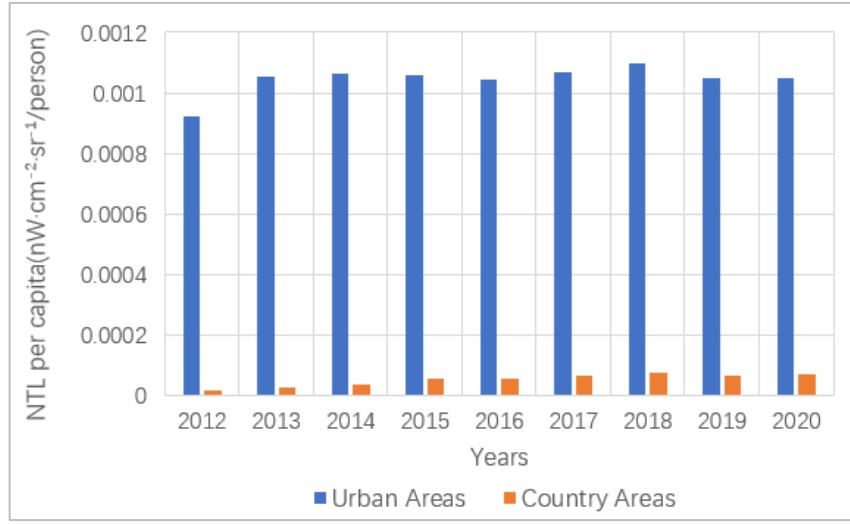

**Figure 10.** NTL per capita of the urban and rural areas in Rwanda from 2012 to 2020.

To generally analyze the spatial patterns of the changes in the nighttime light in Rwanda, we extracted the new lit areas from 2012 to 2020, which are shown in Figure 11. Red areas show the outcomes of power grid expansion, highlighting places that were dark in 2012 but appeared in light in 2020. Overall, the image captures the considerable increase in the spread and coverage of electricity provision since 2012 in Rwanda.

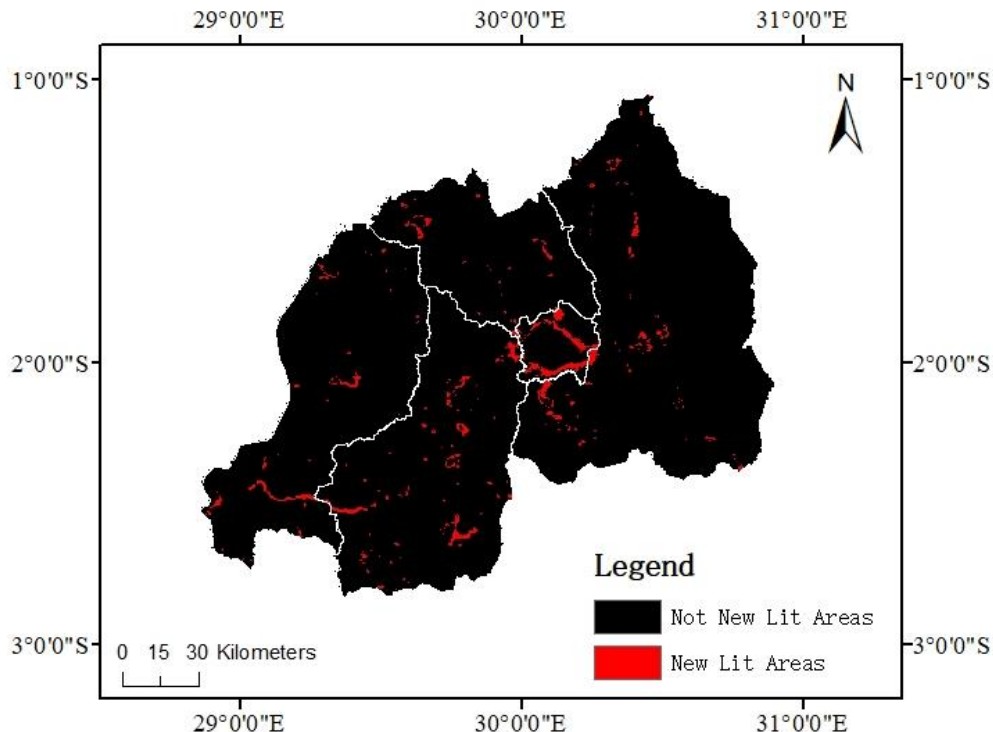

**Figure 11.** The new lit areas in Rwanda from 2012 to 2020.

### 3.2. Grid Access to Settlements

The lit ratio indicates the proportion of the surface area of the lit settlements compared to that of the total settlements. By calculation, we estimated that the lit ratio of the settlements in Rwanda in 2015 was 38.54% and that in 2019 was 25.32%. The results show that the lit ratio in 2019 was even lower than that in 2015, indicating that the construction of electricity infrastructure is lagging behind the expansion of settlements in Rwanda. In order to further analyze the regional disparity in the grid access of settlements, we calculated the lit ratio of each province in Rwanda. The results are as shown in Figure 12.

The following findings were obtained from the results:

(1) The lit ratio of each province in 2019 was lower than that in 2015, indicating that the electricity infrastructure construction in four provinces and Kigali fell behind the expansion of the settlements. The result illustrates that insufficient electricity infrastructure construction is a common phenomenon in Rwanda, which calls for attention;

(2) Notably, the decline in the lit ratio from 2015 to 2019 in Kigali and Eastern Province was much slighter compared to that in the other three provinces. The decrease in the lit ratio in Eastern Province was about 2.1% and that in Kigali was less than 10%. The grid access of the new settlements in these two provinces was more extensive;

(3) The lit ratio of Kigali was much larger than that of the other four provinces, which reveals the serious imbalance in the electrification progress in Rwanda. The capital Kigali has a relatively high electricity infrastructure level, which can basically meet the electricity demand, while there is still a huge electricity gap in the other four provinces. In 2019, the lit ratio of Kigali was approximately 88.44%, while that of the other four provinces was below 20%.

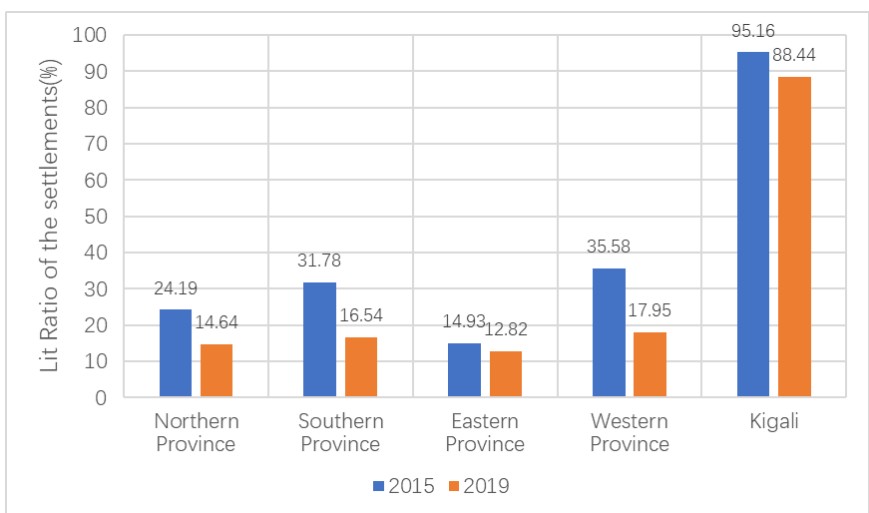

**Figure 12.** The lit ratio of each province in Rwanda in 2015 and 2019.

Following this, we analyzed the impact of the original electricity infrastructure level and the spatial location on the grid access of new settlements.

(1)    The original electricity infrastructure level

The original electricity infrastructure level and the grid access of new settlements are reflected by the initial lit ratio and the lit ratio of new settlements, respectively. Using Equation (6), the $R^2$ of the linear regression between the initial lit ratio (the lit ratio of the settlements in 2015) and the lit ratio of new settlements was 0.695 ($p < 0.05$), showing a strong correlation. The scatter diagram and the linear fitting curve are shown in Figure 13.

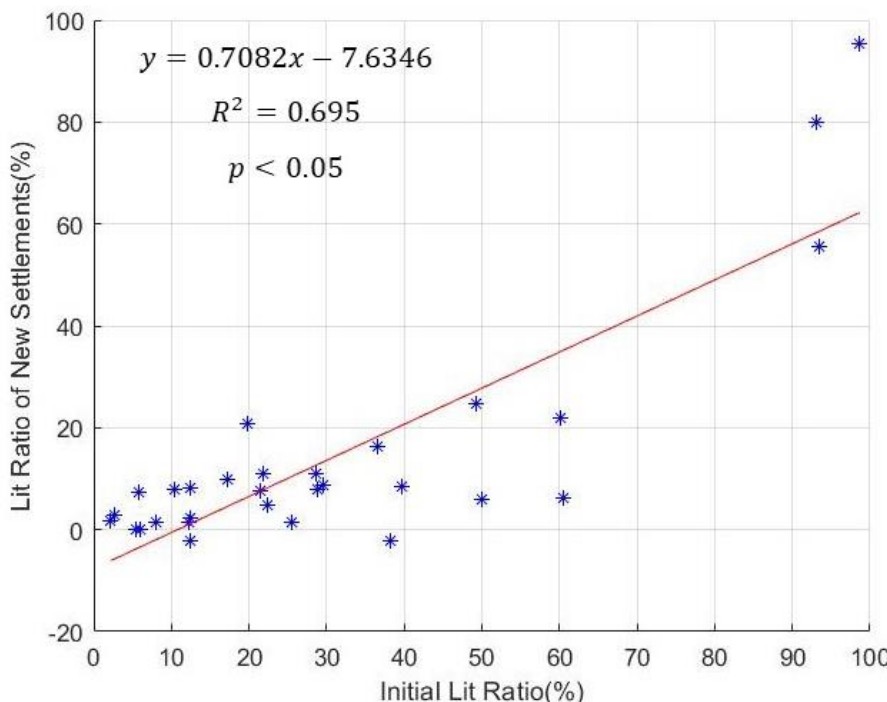

**Figure 13.** The scatter diagram and the linear fitting curve showing the relationship between the initial lit ratio and the lit ratio of the new settlements in 30 districts of Rwanda. In the figure, y denotes the lit ratio of new settlements, and x denotes the initial lit ratio. The $R^2$ of the linear regression is 0.695 ($p < 0.05$).

From the result, we can see that the lit ratio of new settlements increased with the initial lit ratio. Therefore, a region with a higher original electricity infrastructure level can achieve more extensive grid access for its new settlements. To further confirm our finding, we calculated the initial lit ratio (lit ratio of the settlements in 2015) and the lit ratio of the new settlements of each province in Rwanda, and the results are shown in Figure 14.

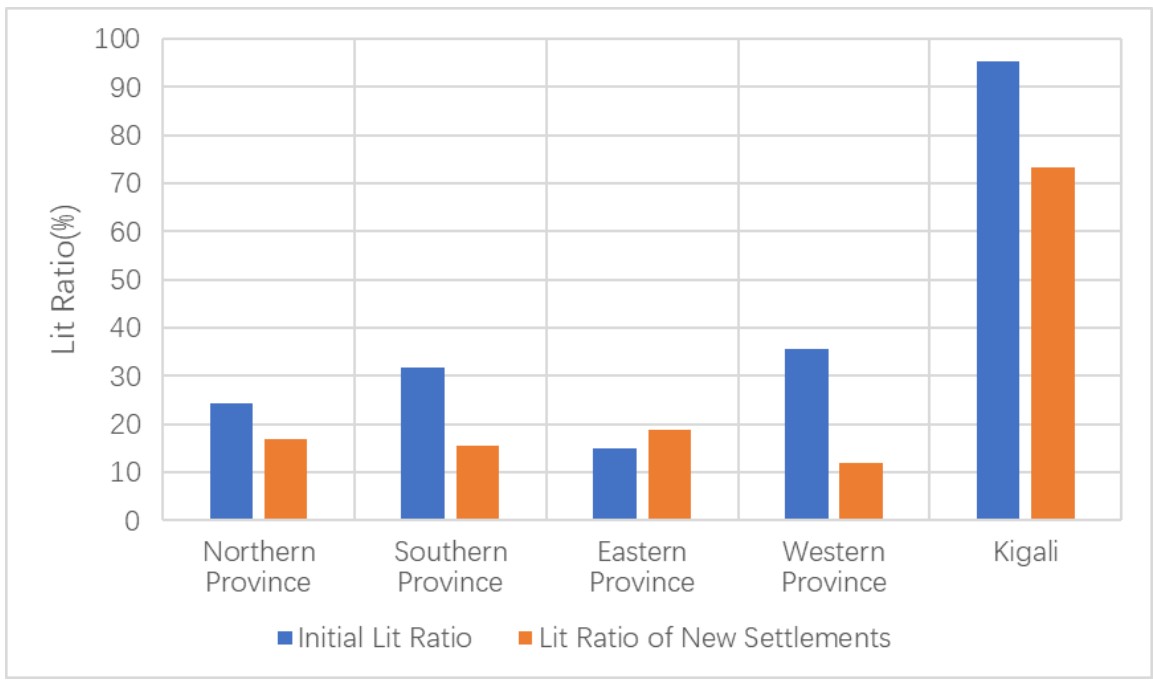

**Figure 14.** The initial lit ratio and the lit ratio of the new settlements from 2015 to 2019 of each province in Rwanda.

As shown in Figure 14, the initial lit ratio of Kigali was much larger than that of the other four provinces, and, correspondingly, its lit ratio of new settlements was also much larger than that of the other four provinces. This is approximately consistent with the finding that the lit ratio of new settlements was positively correlated with the initial lit ratio.

(2) The spatial location

In addition, we analyzed the impact of the spatial location on the grid access of the new settlements. Generally, the electricity infrastructure level in urban areas is higher than that in rural areas [33]. Therefore, we inferred that the cost and difficulty of achieving grid access in new settlements are lower in the urban areas. To confirm our hypothesis, the lit ratio of the new settlements inside and outside the urban boundary was calculated. The lit ratio of the new settlements in Rwanda inside the urban boundary was 21.64%, while that outside the boundary was 2.33%. From the results, it can be seen that the new settlements inside the urban boundary have achieved much more extensive grid access than those outside the boundary, which proves our hypothesis. The results indicate that the spatial location of new settlements has a certain impact on their grid access.

To find the possible impact of the disparity in the spatial location of the new settlements in different provinces on the regional disparity of the lit ratio, we calculated and compared the change rates of the settlements inside and outside the urban boundary at the provincial level, and the results are shown in Figure 15.

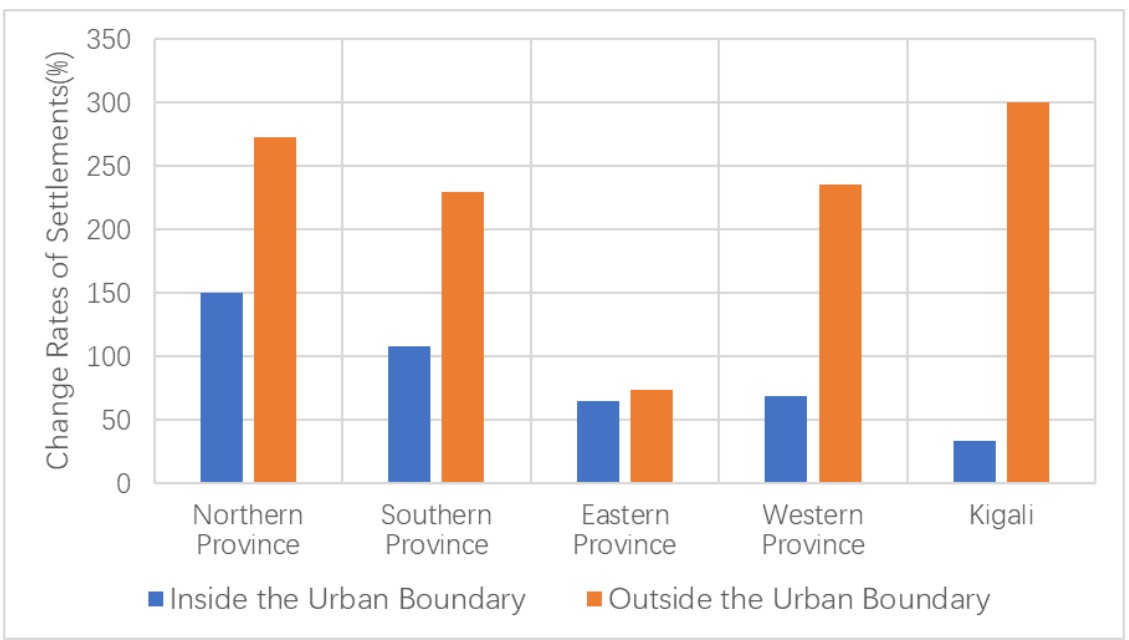

**Figure 15.** The change rates of the settlements inside and outside the urban boundary of each province in Rwanda.

From the results, the following findings were obtained:

(1)  The change rate of the settlements outside the urban boundary was much higher than that inside the boundary, except in Eastern Province, which indicates that most of the new settlements in Rwanda are not in the urban boundary, increasing the difficulty for new settlements to achieve grid access;

(2)  The change rate of the settlements outside the urban boundary in Eastern Province was much lower than that in the other four provinces, which is probably conducive to it achieving extensive grid access for its new settlements;

(3)  The gap between the change rates of the settlements inside and outside the urban boundary in Eastern Province was small, while the change rate of the settlements outside the boundary was much higher than that inside the boundary in the other four provinces.

The above findings may explain why the decline in the lit ratio from 2015 to 2019 in Eastern Province was much slighter compared to that in the other three provinces. Our results have proved that the new settlements inside the urban boundary can achieve much more extensive grid access than those outside the boundary. Compared with the relatively radical expansion of the settlements in the other provinces, the expansion of the settlements outside the urban boundary in Eastern Province was much slower, which indicates that it did not rush into a massive settlement expansion. Therefore, the new settlements in Eastern Province had an advantage in achieving extensive grid access, which illustrates why its lit ratio stayed relatively stable instead of decreasing dramatically from 2015 to 2019.

*3.3. Grid Access Rates on Multi-Spatial Scales*

By dividing the sum of the population with access to grid by the total population, we calculated the grid access rates in Rwanda from 2012 to 2020. The temporal trend in the grid access rates is shown in Figure 16. From the results, it can be seen that the national grid access rate increased from 2012 to 2016 from 14.39% to 18.84%. After 2016, the grid access rate tended to be stable, around 20%.

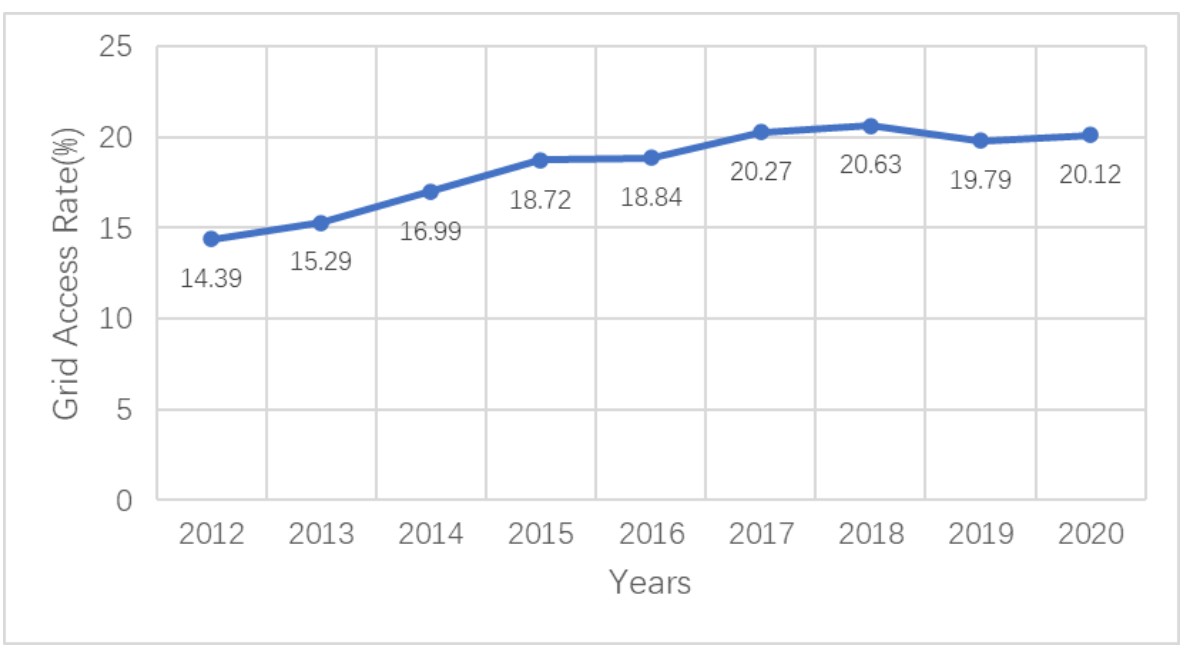

**Figure 16.** Grid access rate of Rwanda from 2012 to 2020.

To compare the grid access in different provinces in Rwanda, we calculated the grid access rates at the provincial level. The temporal trends in the provincial grid access rates from 2012 to 2020 are shown in Figure 17. From the results, it can be seen that the grid access rate of Kigali was much higher than that of the other four provinces, indicating a serious imbalance in grid access among provinces in Rwanda. In 2020, the grid access rate of Kigali reached 90.22%, while that of the other four provinces was only about 10%. While Kigali is almost fully connected to the power grid, the other four provinces still have large populations without a steady supply of electricity.

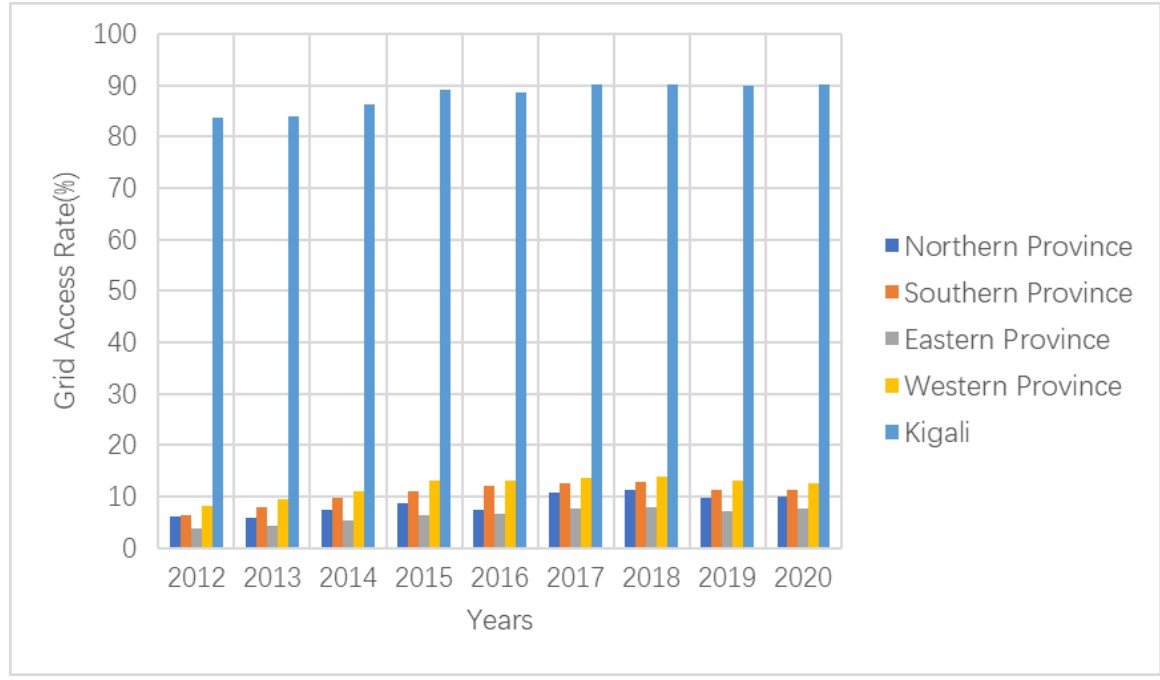

**Figure 17.** The provincial grid access rates in Rwanda from 2012 to 2020.

To understand and compare the grid access condition in the urban and rural areas in Rwanda, we calculated the grid access rates of the urban and rural areas, respectively. The results are shown in Figure 18. From the calculated results, it can be seen that the grid access rates of the urban areas were significantly higher than those of the rural areas. In 2012, the grid access rate of the urban areas was 61.52%, while that of the rural areas was 5.08%. In 2020, the grid access rate of the urban areas was 69.73%, while that of the rural areas was 9.12%. The results show the huge gap between rural and urban grid access in Rwanda.

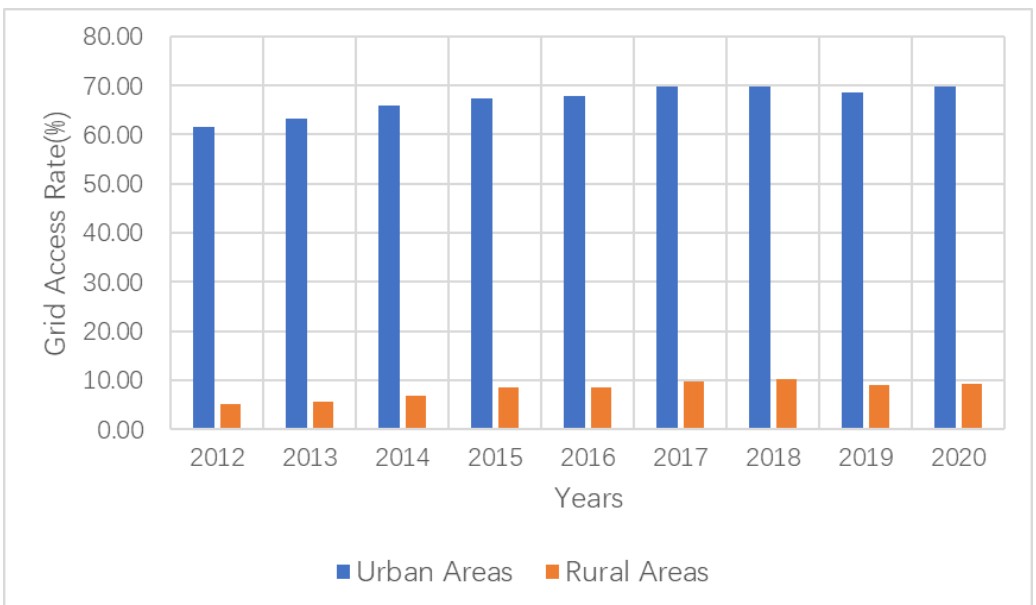

**Figure 18.** The grid access rates of the urban and rural areas in Rwanda from 2012 to 2020.

From the results above, we found that the grid access rates showed similar changing trends on multi-spatial scales: increasing from 2012 to 2016 and staying stable after 2016. To further explore the possible reasons for this trend, we investigated the electrification strategies of Rwanda. The government of Rwanda gave priority to extending the national distribution system across the country and providing consumers with access to grid electricity before 2016 [1], which corresponds to our finding that the grid access rates increased from 2012 to 2016. However, the government has started to promote off-grid access since 2016 and has shifted part of their efforts relating to power grid construction to off-grid expansion [1], which is consistent with the finding that the grid access rates generally tended to be stable after 2016. In Section 4, we further discuss the development of off-grid access in Rwanda.

## 4. Discussion

This study demonstrates one of the many applications of NTL used as a proxy for grid access and highlights the imbalance of electrification progress in Rwanda. It also points out a common phenomenon in Rwanda where electricity infrastructure construction falls behind the expansion of settlements and illustrates the possible influencing factors of power grid expansion. The findings clarify the electrification progress in Rwanda.

### 4.1. The Development of Off-Grid Access in Rwanda

Although grid expansion proves to be a good way to increase electrification, analysis has shown that power grid connections may be economically inefficient in the short-to-medium term for households which use a small amount of electricity [60]. Furthermore, grid expansion is a slow process and may take decades to reach all households [60]. Off-grid technologies have developed significantly in recent years and now present a feasible

alternative to grid access. Off-grid solutions, such as SHS, play a key role in Rwanda's electrification progress [60]. The Rural Electrification Strategy (RES) of Rwanda, published in June 2016, sets out a clear development plan for the off-grid sub-sector. The target for electricity access is for 100% of households to have access through a combination of on-grid and off-grid supply [1].

This study compared the calculated results of the national grid access rates with the official electricity access rate data of Rwanda obtained from the World Bank, shown in Figure 19. From the line chart, it can be seen that the calculated grid access rates from 2012 to 2015 are close to the official data, but the gap between the calculated results and the official data has increased significantly since 2016. In 2020, the calculated grid access rate was 20.12%, and the official electricity access rate was 46.6%, which is almost twice the calculated result.

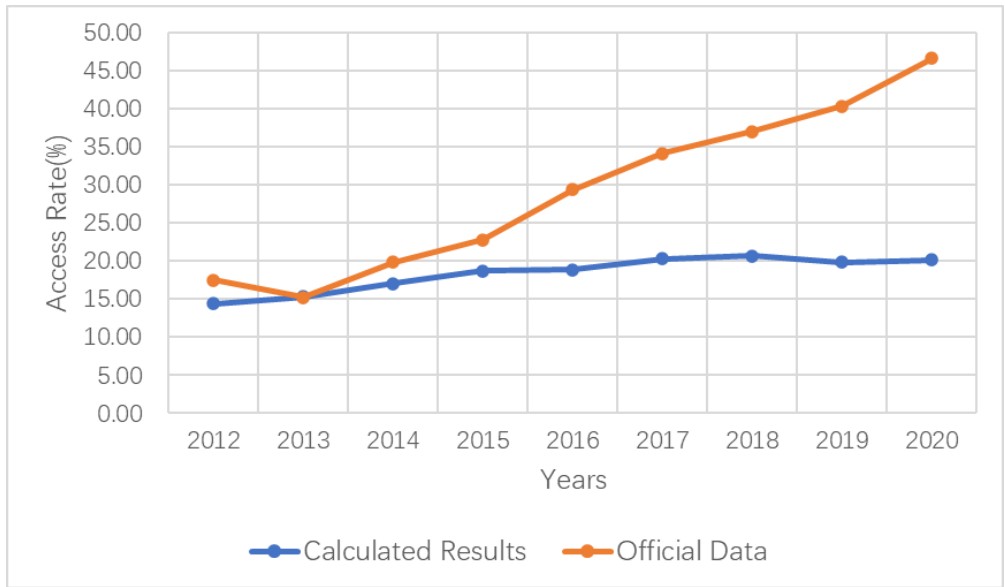

**Figure 19.** Comparison between the calculated national grid access rates and the official electricity access rates data of Rwanda provided by the World Bank from 2012 to 2020.

To find the reasons for this gap, we investigated Rwanda's electrification strategy. The government of Rwanda set a goal to achieve 100% electricity access by 2024 [60]. For the above target to be met, a combination of grid and off-grid access that focuses on the spatial location and electricity consumption level is required instead of the traditional connection to the power grid, which may not be suitable for all households [1]. Off-grid access includes a wide range of technologies, predominantly SHS in Rwanda, which can light an entire house and power appliances such as a television [2]. There is significant scope for off-grid access to provide essential electricity supply for a significant proportion of households in Rwanda.

As clarified at the beginning of Section 2.2, the lit areas extracted in this study were only the areas connected to the grid, while the electricity access rate data provided by the World Bank take both grid access and off-grid access into account. We inferred that this is the reason why our calculated grid access rates were lower than the official electricity access rate data, and the gap related to off-grid access. Additionally, the government of Rwanda has started to promote off-grid access since 2016 [1]. The time point at which the gap appeared exactly matches the time when the government started to vigorously develop off-grid electricity access throughout the country, which further confirms the reason given.

### 4.2. The Spatial Disparity of Off-Grid Access in Rwanda

The DHS provides electricity access rate data at the provincial level in Rwanda, including grid access and off-grid access. Therefore, to further analyze the development of

off-grid access at the provincial level, we compared the calculated grid access rates with the official electricity access rate data of each province in Rwanda. Since the DHS only provides data for a limited number of years, this study only compared the results for 2013, 2015, 2017, and 2020. The results are shown in Figure 20.

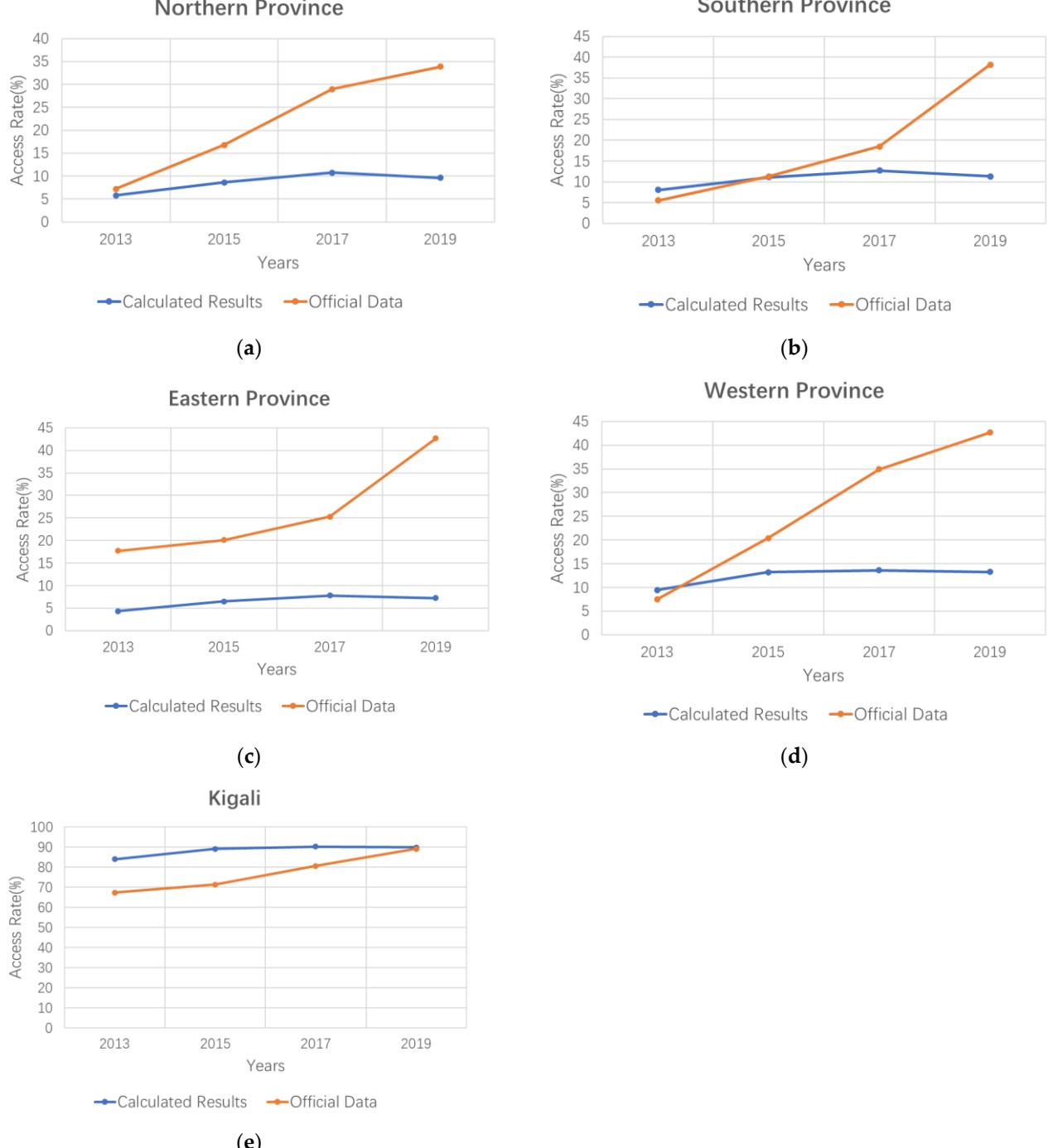

**Figure 20.** Comparison between the calculated grid access rates and the official electricity access rate data in 2013, 2015, 2017, and 2019 in: (**a**) Northern Province; (**b**) Southern Province; (**c**) Eastern Province; (**d**) Western Province; (**e**) Kigali.

As shown in Figure 20, the electricity supply in Kigali was dominated by grid access, while the calculated grid access rates of the other four provinces were much lower than the official electricity access rate data. Therefore, it can be inferred that the other four provinces

had a large proportion of off-grid access to the electricity supply. Furthermore, the gap between the calculated grid access rates and the official electricity access rate data in the other four provinces increased dramatically in recent years, which indicates that off-grid access in the four provinces is growing rapidly.

The DHS also provides electricity access rate data of the urban and rural areas in Rwanda, which can be compared with our calculated grid access rate results. In addition, since the electricity access rate data in urban and rural areas are only provided for a limited number of years, this study only compared the results for 2013, 2015, 2017, and 2020. The results are shown in Figure 21.

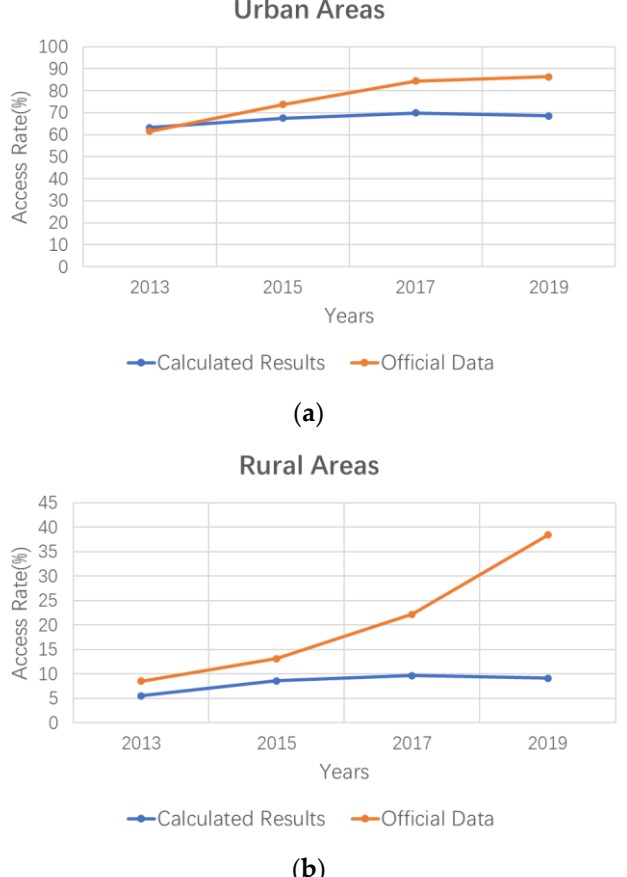

**Figure 21.** Comparison between the calculated grid access rates and the official electricity access rate data in: (**a**) urban areas and (**b**) rural areas.

As shown in Figure 21, urban electricity supply was mainly dominated by grid access all through the years. By contrast, the proportion of grid access for rural electricity supply decreased, while the proportion of off-grid access increased dramatically, which is exactly consistent with the government's policy of vigorously promoting off-grid access in rural areas [1]. These findings suggest that off-grid access is rapidly developing as a major alternative electricity supply in addition to grid access in Rwanda. Especially in rural areas, off-grid access is making up a larger proportion of the total electricity supply than grid access, becoming an indispensable way to obtain electricity. Off-grid access provides a solution for many people who previously had no access to electricity, playing a vital role in dealing with Rwanda's electrification imbalance phenomenon and its efforts to achieve the SDGs.

*4.3. Analysis of Regional Disparity in Electrification*

Previous studies have made lots of efforts to analyze regional inequality using nighttime light satellite imagery, which was mentioned in Section 1. Compared to the previous studies, our study had uniqueness in the following three aspects:

(1) Different study purpose: Our study aimed to track the electrification progress in Rwanda at multi-spatial scales and analyze the disparity in electrification among different provinces as well as between urban and rural areas. Most of the previous studies aimed to develop an index to reflect the disparity within the region;

(2) Different study methods: Our study analyzed the regional disparity in electrification in Rwanda by data comparison instead of by developing an index. For example, we evaluated the disparity in electrification by calculating and comparing the lit ratio and the grid access rate at the provincial level. In this study, it was not suitable or possible to use NLDI like the previous studies mentioned above did to analyze disparity in electrification on multi-spatial scales;

(3) Multi-dimension analysis: Our study analyzed the regional disparity in electrification on multiple dimensions by using NTL imagery and ancillary data. For example, we analyzed the disparity in the electrification of settlements as well as the disparity in grid access rates. The previous studies mentioned above usually only analyzed the regional disparity in one dimension. Our study further expanded the research dimensions of regional disparity and more comprehensively evaluated the imbalance.

*4.4. Limitation and Future Work*

However, some limitations should be noted. As the orbit period of the Suomi NPP is 16 days, different dates of VIIRS data mean different viewing angles [46]. Therefore, the nighttime light derived from VIIRS DNB data shows a strong angular effect, particularly across dense urban centers where NTL radiance at its nadir can be significantly higher than off-nadir observations [44]. Previous studies have explored the effect of different viewing angles on time series [46,61,62]. For example, Li et al. investigated the relationship between viewing angles and the VIIRS DNB radiance of the Suomi NPP satellite in urban areas [61]. Zheng et al. proved that the complicated spatial configuration of highly developed areas, especially areas with high-rise vertical structures, is subject to a stronger angular effect and, thus, greater NTL variation [46].

The angular effect causes uncertainties in the VNP46 product, resulting in high variation in daily NTL data [46]. Temporal composites were proved to be able to reduce the spatiotemporal variation of NTL data and greatly deviated the NTL magnitude from daily observation [46]. When more valid observations are available for generating the composite data, it will be possible to better reduce the NTL variation of daily observation and, thereby, the resulting composite data will be more stable [46]. In this study, we employed the all-angle snow-free layer in the VNP46A4 annual nighttime light composite product. NASA's Black Marble annual NTL composites are generated for multiple-view-angle categories (i.e., near-nadir, off-nadir, and all angles) [44]. The all-angle composites are derived from the greatest number of nights per year compared to either the near-nadir composites or the off-nadir composites [42]. Therefore, by using the all-angle annual composite data, this study mitigated the angular effect to construct more stable time series VIIRS DNB imagery. In the future, the composites of high-quality daily NTL data can be used more for scientific observations.

**5. Conclusions**

Despite having been seriously damaged by the Rwanda Genocide, Rwanda has achieved rapid economic growth through great efforts. With the rapid social and economic development, there is interest in having more spatially explicit and timely data on Rwanda's electrification progress. To satisfy the requirement, we analyzed the spatial pattern of Rwanda's electrification development using annual nighttime light composite imagery.

Through the observation, the following core conclusions were drawn. First, in terms of both the sum of light and grid access, Kigali has an absolute advantage over the other four provinces, which reflects the extreme imbalance in the electrification development of Rwanda. Second, the lit ratio of settlements in 2019 was generally lower than that in 2015, indicating that electricity infrastructure construction lagged behind the expansion of the settlements, which is a common phenomenon in Rwanda. In addition, the original electricity infrastructure level and the spatial location are two influencing factors of grid access in new settlements. Finally, we found that off-grid access is growing rapidly in Rwanda, which is consistent with the government's strategy for promoting off-grid access. The proportion of off-grid access in Rwanda's electricity supply has been increasing dramatically since 2016, especially in its rural areas. In this study, we revealed that the traditional way for estimating electrification using nighttime light imagery has limitations; off-grid access is usually not included in the evaluation of electrification.

Sub-Saharan Africa still faces a lot of difficulties in achieving the SDGs, as there are still many people without electricity access, especially in rural areas. More spatially explicit information on the ongoing electrification progress in sub-Saharan Africa is essential. Nighttime light data will not completely replace the information gathered by statistical survey. Although remote sensing does have its advantages when detailed electricity access rates data are publicly unavailable, it is hard to accurately survey the electrification status based solely on nighttime light imagery. Yet inferences drawn from nighttime light imagery can help us to better understand the electrification progress in sub-Saharan Africa and provide guidance for it to achieve the SDGs. If nighttime light imagery gains higher spatial resolution and wider radiometric range, more details on the electrification progress will be revealed. Our future work will focus on the application of nighttime light imagery with higher spatial resolution in evaluating the electrification status in some countries where the related data are publicly unavailable.

**Author Contributions:** Conceptualization, X.L.; methodology, Y.R.; analysis and investigation, Y.R. and X.L.; writing (original draft preparation), Y.R.; writing (review and editing), X.L., Y.R. and W.A.B. All authors have read and agreed to the published version of the manuscript.

**Funding:** This work was supported by the National Key R&D Program of China under Grant 2019YFE0126800 and the Project of Innovation and Entrepreneurship Training of National Undergraduate of Wuhan University (GeoAI Special Project) under Grant S202110486218.

**Data Availability Statement:** Not applicable.

**Acknowledgments:** We give thanks to the editors and anonymous reviewers for their valuable comments to improve our manuscript.

**Conflicts of Interest:** The authors declare no conflict of interest.

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
