# Peer review of "Tracking Spatiotemporal Patterns of Rwanda’s Electrification Using Multi-Temporal VIIRS Nighttime Light Imagery"

_remotesensing, doi:10.3390/rs14174397_

Round 1

Reviewer 1 Report

The paper overall makes sense , however a quick search of similarly relevant papers and looking at the references made me wonder why two papers were not included:

Elvidge, C. D., Baugh, K. E., Anderson, S. J., Sutton, P. C., & Ghosh, T. (2012). The Night Light Development Index (NLDI): a spatially explicit measure of human development from satellite data. Social Geography7(1), 23-35.

- And Papers like: Singhal, A., Sahu, S., Chattopadhyay, S., Mukherjee, A., & Bhanja, S. N. (2020). Using night time lights to find regional inequality in India and its relationship with economic development. PloS one15(11), e0241907.

If the authors can explain how their methods are different / complement the efforts mentioned above - this would be extremely helpful to understand the uniqueness of the research. 

This is my only "major" revision - help me understand if you have categorized/cited similar papers in your Introduction or your Methods sections...

Thank you!

Author Response

Dear Reviewer:

Please see the attached file as the response.

Reviewer 2 Report

This paper :” Tracking spatiotemporal patterns of Rwanda’s electrification using multi-temporal VIIRS nighttime light imagery” by Yuanxi Ru et al. used the VIIRS Day Night Band imageries to detect the electrification progress at different regions of Rwanda from 2012 to 2020. They associated the DNB light with the social and economic development. This study looked at the lit ratio, the grid expansion and settlement relations, the disparity between rural and urban area, among 4 provinces and the capital. The authors used NASA black marble products. The DNB radiance are all view-angle datasets. This study provides an excellent example how the VIIRS DNB can be used to detect human activities and development. The study is well designed, and the reference data are well selected from different social/economic/satellite datasets. The logic is sound, and the paper is well written.  I recommend this paper published at Remote Sensing with minor modifications.

Minor Comments:

1.)    Line 210 NTLNTL, repeated NTL.

2.)    The NASA composite black marble data sets include all-view-angle DNB radiance. The DNB radiance level changes with view angle significantly. Given the VIIRS orbital periods of 16 days and the annual product the authors used, how does this dependency of the radiance on the view angle affect the time series?  

3.)    It would be better to give a few sentences on the black mark data product data quality and how it is compared with NOAA NCEI DNB product: https://ngdc.noaa.gov/eog/viirs/download_ut_mos.html?  

Author Response

(The authors gave the same response as above.)

Round 2

Reviewer 1 Report

Thanks for making the adjustments - paper looks good now. Would just run spell check / grammar check.